# Age-Related Changes in Lipid and Glucose Levels Associated with Drug Use and Mortality: An Observational Study

**DOI:** 10.3390/jpm12020280

**Published:** 2022-02-14

**Authors:** Rene Markovič, Vladimir Grubelnik, Helena Blažun Vošner, Peter Kokol, Matej Završnik, Karmen Janša, Marjeta Zupet, Jernej Završnik, Marko Marhl

**Affiliations:** 1Faculty of Natural Sciences and Mathematics, University of Maribor, 2000 Maribor, Slovenia; rene.markovic@um.si; 2Faculty of Electrical Engineering and Computer Science, University of Maribor, 2000 Maribor, Slovenia; vlado.grubelnik@um.si (V.G.); peter.kokol@um.si (P.K.); 3Community Healthcare Center Dr. Adolf Drolc Maribor, 2000 Maribor, Slovenia; helena.blazun@zd-mb.si; 4Faculty of Health and Social Sciences, 2380 Slovenj Gradec, Slovenia; 5Alma Mater Europaea—ECM, 2000 Maribor, Slovenia; 6Department of Endocrinology and Diabetology, University Medical Center Maribor, Ljubljanska ulica 5, 2000 Maribor, Slovenia; matej.zavrsnik1@gmail.com; 7The Health Insurance Institute of Slovenia, Miklošičeva cesta 24, 1507 Ljubljana, Slovenia; karmen.jansa@zzzs.si (K.J.); marjeta.zupet@zzzs.si (M.Z.); 8Science and Research Center Koper, 6000 Koper, Slovenia; 9Faculty of Education, University of Maribor, 2000 Maribor, Slovenia; 10Faculty of Medicine, University of Maribor, 2000 Maribor, Slovenia

**Keywords:** diabetes, metabolic syndrome, hematological data, aging

## Abstract

Background: The pathogenesis of type 2 diabetes mellitus is complex and still unclear in some details. The main feature of diabetes mellitus is high serum glucose, and the question arises of whether there are other statistically observable dysregulations in laboratory measurements before the state of hyperglycemia becomes severe. In the present study, we aim to examine glucose and lipid profiles in the context of age, sex, medication use, and mortality. Methods: We conducted an observational study by analyzing laboratory data from 506,083 anonymized laboratory tests from 63,606 different patients performed by a regional laboratory in Slovenia between 2008 and 2019. Laboratory data-based results were evaluated in the context of medication use and mortality. The medication use database contains anonymized records of 1,632,441 patients from 2013 to 2018, and mortality data were obtained for the entire Slovenian population. Results: We show that the highest percentage of the population with elevated glucose levels occurs approximately 20 years later than the highest percentage with lipid dysregulation. Remarkably, two distinct inflection points were observed in these laboratory results. The first inflection point occurs at ages 55 to 59 years, corresponding to the greatest increase in medication use, and the second coincides with the sharp increase in mortality at ages 75 to 79 years. Conclusions: Our results suggest that medications and mortality are important factors affecting population statistics and must be considered when studying metabolic disorders such as dyslipidemia and hyperglycemia using laboratory data.

## 1. Introduction

Diabetes is a complex, multifactorial metabolic disease that is related to other age-related diseases, particularly cardiovascular disease (CVD), neurodegenerative disease (NDD), and cancer. Both the number of cases and the prevalence of diabetes are steadily and continuously increasing. Globally, the number of people with diabetes increased from 108 million to 422 million and to 463 million between 1980, 2014, and 2019 [1,2]. The global prevalence (age-standardized) of diabetes in adults has almost doubled since 1980, increasing from 4.7% to 8.5% in 2014 and 9.3% in 2019 [3]. In 2019, an estimated 4.2 million deaths were attributable to diabetes and its complications [2]. High blood glucose causes another 2.2 million deaths by increasing the risk of developing CVD and other diseases. Nearly half of all deaths attributable to high blood glucose occur before the age of 70 years. According to the WHO, diabetes was the seventh leading cause of death in the United States in 2017 [1,4] and is currently the eighth leading cause of death worldwide [5]. Diabetes also represents a tremendous financial burden: it is estimated that global healthcare expenditure for the treatment of diabetes was $760 billion in 2019 and will increase to $845 billion by 2045 [3].

Most people with diabetes are affected by type 2 diabetes mellitus (T2DM), which accounts for approximately 90% of all cases [6]. T2DM is an increasingly prevalent, near epidemic, non-communicable disease associated with genetic and non-genetic risk factors such as overweight, obesity, diet, stress, and sedentary lifestyle. In a world where the consumption of processed foods and added sugars, especially fructose, is increasing, there is growing concern about the epidemiological spread of T2DM [7,8,9,10]. Extensive research has been conducted in recent decades to gain a better understanding of the pathogenesis of T2DM. Prevention and early diagnosis of T2DM, especially the detection of prediabetic conditions, particularly metabolic syndrome (MetS), are of particular importance. As an aside, biomarkers used to predict MetS are already more effective in detecting patients who have or will develop T2DM than those with CVD complications [11]. However, the increasing prevalence of diabetes worldwide suggests that even more effective biomarkers are needed for the early diagnosis of metabolic disorders leading to MetS and T2DM.

Regarding early biomarkers and risk factors, obesity and dyslipidemia have been associated with MetS [12,13]. Weight gain and obesity are important risk factors for various diseases, from insulin resistance and T2DM to atherosclerosis and nonalcoholic fatty liver disease [14,15]. Lipid disorders in MetS and T2DM are particularly associated with elevated triglycerides (TG), low levels of high-density cholesterol (HDL), and often elevated total cholesterol. This combination, accompanied by small dense LDL (sLDL), is known as “lipid triad” [15,16] or diabetic dyslipidemia [16,17], which is particularly common in patients older than 65 years [18]. How exactly dyslipidemia is related to hyperglycemia and T2DM is not clear. Several experiments have been conducted to better understand the relationship between dyslipidemia and glycemic disorders. For example, in morbid obesity, elevated fasting free fatty acids (FFA) and impaired FFA suppression are associated with hypertriglyceridemia [19,20,21]. High levels of FFA have been found to mediate insulin resistance in target tissues [22,23,24,25,26,27,28]. The lack of suppression of FFA is associated with inhibition of carbohydrate oxidation and glycogen synthesis in muscle during hyperinsulinemia [29], reduced insulin clearance by the liver [30], and elevated VLDL production [31]. The mechanisms of the relationship between insulin resistance and dyslipidemia are complex and were recently reviewed in [32,33].

In T2DM, elevated blood glucose appears to be a late manifestation of impaired glucose regulation, and other indicators could be used as biomarkers for T2DM, including disturbances in lipid metabolism [34]. Several studies suggest that pediatric metabolic abnormalities predict MetS and T2DM in adulthood [35]. There is also evidence that lipid metabolic abnormalities may precede T2DM by several years [16,36,37,38,39,40,41]. However, the various components of diabetic dyslipidemia are not isolated abnormalities but are closely related to a broader spectrum of metabolism [17]. Recent research has also shown that metabolism and epigenetics are associated with a complex relationship between metabolism and DNA methylation and histone modification, and that their dysregulation leads to the onset of T2DM [42].

Several statistical studies have also been conducted to answer this complex question of how metabolic disorders, and in particular dyslipidemia, are associated with high blood glucose levels and T2DM. In a population-based cross-sectional study involving only healthy and drug-free subjects, age-related trends in serum glucose and lipid profiles were analyzed [38]. Leao et al. [43] conducted a longitudinal and retrospective study and investigated the association between lipid-related parameters and gender. Natali el al. [44] performed a prospective, multicenter, cross-sectional observational study in nondiabetic subjects and investigated the association between serum lipids and changes in insulin secretion or clearance in nondiabetic subjects. Huang et al. [45] analyzed fasting blood glucose levels in individuals 18 years or older from 2007 to 2015 as part of a cross-sectional study. Schubert et al. [39] analyzed data from the Fels Longitudinal Study of 269 healthy participants and examined whether age-related deterioration in lipid profiles and serum glucose levels is primarily influenced by body composition and lifestyle or by other aspects of aging.

Some epidemiological studies have also examined medical treatment protocols and the corresponding effects on lipid and/or glucose metabolism. Toth et al. [46], in a retrospective data analysis using medical and pharmaceutical records, laboratory results for LDL, HDL, and TG, and enrollment information from 186,000 patients, examined the treatment patterns of patients with mixed dyslipidemia and CVD risk factors. Pettersson et al. [47] conducted a retrospective longitudinal study focusing on the prevalence of dyslipidemia and achievement of target lipid levels in patients treated with lipid-modifying therapy. Hopstock et al. [48] examined longitudinal and secular trends in total cholesterol levels and the effect of lipid-lowering agents. Tuppin et al. [49] analyzed treatment frequency of antihypertensive, lipid-lowering, and antidiabetic medications in 58 million individuals, excluding lipid profiles and serum glucose levels.

Previous epidemiologic studies have also examined associations between mortality and medication use or between mortality and lipid profiles and serum glucose levels. For example, Yu et al. [50] conducted a cohort study of 6941 hypertensive patients aged 65 years or older who were not treated with lipid-lowering agents and investigated the potential of lipid profiles as an effective predictor of all-cause mortality in elderly hypertensive patients. Yi et al. [51,52] examined the association between total cholesterol level as well as fasting glucose level and all-cause mortality, but participants’ medication use was not considered or used as an exclusion criterion. AresI et al. [53] examined the risk of death in adults by categories of impaired glucose metabolism after an 18-year follow-up period in a population-based, prospective cohort study. Gardette et al. [54] analyzed National Registry of Myocardial Infarction data from a prospective population-based cohort study with the aim of comparing 10-year all-cause mortality according to initial dyslipidemia status and exposure to lipid-lowering agents. Ko et al. [55] examined the association between statin therapy and 3-year mortality risk in a retrospective cohort study of 396,077 patients aged 66 years and older, using interlinked administrative healthcare databases, excluding lipid profiles and serum glucose levels. Duncan et al. [56] examined the effects of medication use on serum lipid levels and mortality in a 35-year longitudinal study of 3875 participants. As this overview shows, although much knowledge and information has been gained in the past, new questions and issues have also emerged or have not yet been fully explored.

In the present study, we aim to integrate perspectives on lipid and glucose profiles in the context of age, sex, medication use, and mortality. First, we analyze laboratory data to find evidence of dysregulations in lipid and glucose metabolism and to determine how the dynamics depend on sex and age. The focus is on whether some disorders occur earlier in life than others and to what extent this depends on sex. Second, we are investigating whether and to what extent laboratory data are affected by medication use, and third, we are interested in how the statistics of lipid and glucose measurements are affected by mortality, particularly in older age groups. The data for laboratory tests, medication use, and mortality were obtained from three separate datasets and do not allow for a longitudinal study, but the analysis allows us to clearly demonstrate that lipid and glucose profiles in population statistics are qualitatively altered by medication treatment and mortality. This underscores the importance of providing an overall picture when compiling statistics on specific laboratory data that reflect a wide range of all these parameters, including drug treatment, mortality, and, in perspective, probably many other factors such as lifestyle, dietary habits, and physical activity.

## 2. Materials and Methods

This section provides a detailed description of the datasets used, data preparation, data analysis, and statistical metrics.

### 2.1. Data Description and Demographic Features

The Community Healthcare Center Dr. Adolf Drolc in Maribor, Slovenia (CHDAD) is the second largest healthcare center in Slovenia with more than 200,000 inhabitants and an area of 750 square kilometers. It has collected anonymized data from laboratory analysis of blood tests. We used these data collected in the period between 1 January 2008 and 2 March 2019. For this period, the total number of anonymized laboratory tests was 30,314,455, from 248,363 different patients. Each patient was assigned a unique randomized identifier. This identifier allowed us to protect patients’ personal information while distinguishing patient-specific test results. For our study, we focused only on laboratory results for serum levels of glucose (Gluc), total cholesterol (Chol), low-density cholesterol (LDL), high-density cholesterol (HDL), and triglycerides (TG). The use of data related only to these five laboratory blood tests reduced the original dataset to 506,083 test results, or 2.4% of the total dataset of 63,606 different patients, or 25.6% of all patients in the CHDAD dataset. All results presented in the Results section that relate to the CHDAD dataset are based on analysis of this subset of the CHDAD dataset. For a given patient’s laboratory test result, we also used information about the patient’s gender and age when the test was performed. Based on the age of the patients, we divided the patients into age groups with an age interval of 5 years (i.e., 20–24, 25–29, 30–34, etc.). Because the data were collected over an 11-year period, patients may have fallen into different age groups and there may be multiple test results for a patient within an age group. All blood samples and laboratory procedures were performed according to the relevant protocols for registered laboratories in Slovenia. For a detailed description of the CHDAD dataset used, see Appendix A.

The evaluation of individual test results in terms of normal or abnormal values was based on the reference values used at the Community Healthcare Center Dr. Adolf Drolc in Maribor, which can be found on its website [57]. Normal values for laboratory tests range from 3.6–6.1 mmol/L for serum glucose level, 4.0–5.0 mmol/L for serum cholesterol level, 2.0–3.0 mmol/L for serum LDL level, 0.6–1.7 mmol/L for serum triglyceride level, and above 1.3 mmol/L for serum HDL level.

In addition to the database provided by CHDAD, we used a separate database from the Health Insurance Institute of Slovenia (HIIS). HIIS provided us with an aggregated dataset of anonymized entries for medications dispensed by 1,632,441 different patients in Slovenia between 2013 and 2018. Medications were classified according to the Anatomical Therapeutic Chemical (ATC) classification scheme, the accepted classification system for medicines maintained by the World Health Organization (WHO). The WHO assigns ATC codes to all active ingredients in drugs based on the therapeutic indication of the drug. The present study focuses on drugs classified as drugs for diabetes, DFD, (ATC A10), and lipid- modifying agents, LMA, (ATC C10). After selecting patients from the HIIS database who had received either of these two types of drugs, 392,171 different patients remained (24% of all patients listed in the dataset). In addition, the number of patients who had received lipid-altering drugs and drugs for diabetes was 342,499 and 139,676, respectively. Because we divided patients into age groups, patients may reappear in different age groups. A detailed description of the HIIS dataset for our analysis is provided in Appendix A.

Finally, we used mortality data obtained from the website of the Statistical Office of the Republic of Slovenia [58]. The data obtained indicate the number of deceased within a given age group. The age groups for this dataset are fixed and cannot be adjusted. From the website we downloaded the annual mortality data for 2018.

### 2.2. Data Preparation and Analysis

Data analysis was performed in two steps. First, we checked the data for duplicates and deleted them. In the CHDAD database, this procedure reduced our records to 77,289 different patients. In the second step, special attention was paid to entries that could not be directly converted into numerical values. Such entries were subjected to additional data cleaning steps (removal of all spaces, standardization of the decimal separator, removal of letters, units of measurement, and other non-numeric symbols). If numerical information could still not be extracted from the entries after this, they were deleted. In addition, entries belonging to patients younger than 20 years were removed. After this procedure, the records of 13,683 different patients, or 21.5% of the total dataset, were removed, leaving the remaining test results of 63,606 different patients available for further analysis. From the HIIS dataset, we had patient-specific information on age, sex, and ATC code of prescribed medications. No data were removed during data cleaning (removal of duplicates and missing values), allowing us to use 100% of the dataset for further analysis. The dataset contained 97,404,355 entries belonging to 1,714,790 different patients. After removing 470,026 records belonging to 82,349 patients younger than 20 years of age, 96,934,329 records with drug prescriptions from 1,632,441 different patients remained. These records formed the HIIS dataset used in this study.

In addition, we addressed the issue of multiple laboratory test results that a given patient had in the time window of a given age group. To avoid a bias in favor of patients with multiple measurements, we calculated for all these patients with multiple measurements the average value of all measurements that these patients had in the time window of the age group under consideration. In this way, the data were unambiguously assigned to a particular patient, so that these values were either the single measurement or the average of multiple measurements in the period of the age group. 

To determine whether the differences in the distribution of measured values between two subpopulations, male and female patients, were significant, we performed the Mann–Whitney U test [59]. To determine the statistical significance between any two values (e.g., the proportion of male and female patients of a certain age with elevated glucose levels), we used the chi-square statistic [60].

According to laboratory data (CHDAD database) considering serum levels of glucose (Gluc), cholesterol (Chol), low-density lipoproteins (LDL), high-density lipoproteins (HDL), and triglycerides (TG), we included entries for drugs for diabetes, DFD, (ATC A10), and lipid-modifying agents, LMA, (ATC C10) from the drug database. We focused on medication use to determine the proportion of patients of a given age who had taken a given medication at least once. To assess how prevalent the use of a drug was in a given age group, we defined the proportion as the ratio of the number of patients who had taken a given drug at least once to the number of all patients in the same age group. We were also interested in the rate at which the use of a drug category changed between two successive age groups. To assess this, we calculated the rate of increase in drug use (*RIDU*), defined as follows:(1)$$RIDU_{drug}\left(age\;group\right)=100\;\frac{p_{drug}\left(age\;group+1\right)-p_{drug}\left(age\;group\right)}{p_{drug}\left(age\;group\right)},$$
where $p_{drug}\left(age\;group\right)$ is the proportion of patients in a given age group who have taken a given type of drug at least once, $p_{drug}\left(age\;group+1\right)$ the proportion of patients in the next higher age group who have taken a given type of drug at least once. Thus, $RIDU_{drug}\left(age\;group\right)$ represents the rate at which a particular type of drug increased in a particular age group. Negative values of the variable $RIDU_{drug}\left(age\;group\right)$ result when there is a declining trend in drug use between two successive age groups. Positive values, on the other hand, indicate an increasing trend in drug use between two consecutive age groups.

Finally, we also approximated some trends in laboratory measurements, and used fitting procedures. For power-law fitting we used the following equation:(2)$$y_{P}=a+b\;{\left(AGE-AGE_{0}\right)}^{c},$$
where *AGE* denotes the age group, $AGE_{0}$ corresponds to the youngest age group considered, *a* is the estimated value for the lowest age group, *b* is the scaling factor, and *c* is the scaling order. For the linear fit, we used the following equation:(3)$$y_{L}=k\left(AGE-AGE_{0}\right)+y_{0},$$
where *k* is the slope of the linear function and $y_{0}$ is the estimated value for the youngest age group.

All analyses of the datasets were performed using Python 3.7 [61] in combination with the Python packages Matplotlib [62], SciPy [63], and Pandas [64].

## 3. Results

### 3.1. Lipid and Glucose Levels in Different Age Groups

The proportion of patients who had elevated serum cholesterol, LDL, TG, and glucose levels and low HDL levels is shown in Figure 1. In general, the proportion of patients with elevated blood cholesterol was higher than the proportion of patients with elevated serum glucose. For example, 55.7% of patients in the 40–44 age group had elevated cholesterol levels compared with 9.7% of patients in the same age group whose glucose levels were above normal. Our results also suggest age-specific trends. The proportion of patients with high serum cholesterol levels increased up to the 55–59 age group and decreased thereafter. The proportion of patients with higher-than-normal glucose levels increased until the 75–79 age group and then began to decline. This decline does not mean that patients were becoming healthier, but indicates that patient survival is associated with normal glucose and lipid levels.

The peak of the highest proportion of serum glucose above normal occurred about 20 years after the peak of the highest proportion of serum cholesterol above normal. The proportion of patients with inadequate serum HDL levels varied in younger age groups but remained approximately constant in patients older than 35 years. We also observed that the proportion of patients with elevated TG levels increased up to the age of 55–59 years and started to decrease monotonically after the age of 65–69 years. It should be noted that all these results represent only a snapshot of population data and not a longitudinal view of individuals. We lack data on causal relationships, and we cannot say, for example, that an individual first developed dyslipidemia and then hyperlipidemia, but statistically the highest percentage of individuals with hyperglycemia in the population was observed about 20 years after the peak in the percentage of individuals with hyperlipidemia.

For more information on the sex- and age-specific distribution of serum levels, see the Appendix A. The Appendix A contains additional results regarding differences in laboratory test results, particularly in male and female patients in different age groups. In summary, for serum cholesterol levels, a higher proportion of male patients had elevated serum cholesterol and LDL levels than female patients up to age group 45–49, and the proportion of elevated serum cholesterol and LDL levels was higher in female patients from this age group. In addition, the peak proportion of patients with elevated levels was higher in female patients than in male patients (see Appendix A). The proportion of male patients with elevated serum cholesterol and LDL levels began to decline between 45 and 49 years of age. In women, the proportion started to decrease a decade later (age group 55–59 years). The proportion of male patients with normal serum HDL levels was significantly lower than the proportion of female patients with normal serum HDL levels in all age groups. For serum glucose levels, the proportion of male patients with elevated glucose levels was higher than that of female patients in all age groups. A phase diagram showing the correlation between mean serum glucose levels and mean serum cholesterol levels for different age groups, separately for male and female patients, is shown in Figure 2.

As can be seen in Figure 2, average cholesterol levels increased more steeply than glucose levels, and peaked in the 50–59 age group. In this age group, average serum cholesterol levels began to decline, while average serum glucose levels continued to increase until the 70–79 age group. Thereafter, both mean serum cholesterol and glucose levels began to decline. We observed a shift in the progression when we considered the average cholesterol and glucose levels separately for male and female patients. On average, the course in females was characterized by lower glucose levels but higher cholesterol levels than in males. Consequently, the female course reached the highest serum cholesterol levels a decade later than the male course. Moreover, average serum cholesterol levels never returned to normal in women but remained elevated. In contrast, we observed that the course in male patients returned to normal serum cholesterol levels.

The laboratory test results show that lipid and glucose metabolic disorders increased with age, with a shift in the time course of the disorders between male and female patients. In male patients, the course shifted toward elevated glucose levels. In female patients, the course shifted toward elevated cholesterol levels. The peak proportion of patients with elevated serum cholesterol preceded the peak proportion of patients with elevated glucose by approximately 20 years (Figure 1 and Figure 2). In Appendix A, we have additionally reported the results regarding the proportion of patients with abnormal test results, and the mean and 95% confidence interval of the results. The results are given for individual age groups and separately for male and female patients.

### 3.2. Drug Use in Different Age Groups

We place the results presented above in a broader context. In particular, the measured laboratory data are influenced by many different factors, one of which is drug use. It is difficult to determine explicitly the effect of drugs on the measured data, but certainly the measured “normal” lipid levels in subjects taking lipid-lowering drugs cannot be treated equivalently to those with normal levels without drugs. To assess the influence of medication on the measured values, more comprehensive data with additional information on the patients’ medication use would be required. We do not have this additional information for individual patients. Therefore, we only observed the population cohort to get an idea of the extent and age at which medication use was most prevalent in the population.

For the entire Slovenian population, using the HIIS database (see Section 2), we analyzed the proportion of patients of a given age who were prescribed drugs classified as lipid-modifying agents, LMA (ATC C10), and drugs for diabetes, DFD (ATC A10). For a complete list of all drugs included under LMA and DFD, with the corresponding total frequency of prescriptions, see Appendix A. The prevalence of drug use in the population of a given age and sex is shown in Figure 3.

Figure 3A shows the proportion of all patients who were prescribed either with an LMA or a DFD, or both together (LMA and DFD); for gender-specific prescription of LMA and DFD see Appendix A. Prescription of LMA was more dominant compared with DFD, reaching nearly 45% compared with 20%. The percentage of the population taking medications increased until the 75–79 age group and then began to decline. This decline may indicate that measured lipid and glucose levels in elderly patients are influenced by medication use, among other factors. It appears that lipid and glucose levels initially increased, and medication use was a response to the elevated serum lipid and glucose levels. In Figure 1, we see that serum lipid levels rose earlier and to a much greater extent than glucose levels in the adult population, so LMAs were also used earlier and to a greater extent than DFDs. However, the age-related percentages for use of LMAs and DFDs do not directly correlate with the observed lipid profiles and serum glucose levels in Figure 1, suggesting that drug prescribing is not based on laboratory data alone but that other factors must also be considered. Interestingly, in the 55–59 age group, where the percentage of patients with dyslipidemia was about 75%, the LMA administration was still low at about 20%. This could be partly related to the fact that the standards for the use of antilipemic drugs are not as strictly defined as for drugs for diabetes. Physicians in Slovenia are advised to prescribe the antilipemic drugs when there is an overall risk of cardiovascular disease; dyslipidemia alone, without other risks, is not a reason to prescribe LMA.

To better assess the presence of the two types of medications in the population of a given age and sex, we calculated the LMA/DFD ratio for the total population and separately for male and female patients. The results are shown in Figure 3B. DFD medication was more prevalent in younger patients. This prevalence was sex-specific. The ratio of the proportion of patients receiving LMA medication to the proportion of patients receiving DFD increased until the 55–59 age group and remained at a plateau until the 75–79 age group, when it began to decline. However, these trends were gender-specific. In men, the plateau phase tended to be reached earlier and lasted longer than in female patients. In female patients, the LMA/DFD ratio began to decline after the peak, initially more slowly until the 75–79 age group and more rapidly thereafter. It is also noteworthy that the trend for the total population (black line) approached the trend for females (red line) after the age of 75–79 years, indicating the predominance of female patients in this oldest age group.

The results shown in Figure 3C,D are for diabetic patients and patients with dyslipidemia, respectively. Figure 3C shows how the proportion of diabetics taking LMA in addition to DFD, expressed as (LMA and DFD)/DFD, changed with age. It should be noted that approximately 80% of diabetics between the ages of 60 and 80 also took LMA. The proportion of male diabetics who were also prescribed with medication to regulate serum lipid levels increased earlier than in female patients. However, in female patients, it is observed that the proportion increased much more after the age of 50. A relatively lower proportion was observed in younger patients (under 50 years of age) and then again in older patients (over 80 years of age). The results suggest that younger patients receiving DFD are in most cases not concomitantly treated with LMA. Younger patients were also found to have a higher proportion of DFD compared to LMA (see Figure 3B), which could be associated with type 1 diabetes. At the same time, this ratio was particularly pronounced in women aged 25 to 40 years compared with men. It is likely that this is partly related to gestational diabetes. In elderly patients (80 years and older), the lower ratio of (LMA and DFD)/DFD is not easily explained. This could be due to mortality and discontinuation of prescribed medications due to other diseases.

Finally, Figure 3D shows the extent to which patients with dyslipidemia used both types of drugs (LMA and DFD) according to their age and sex. Compared with Figure 3C, the ratio (LMA and DFD)/LMA is not as high as the ratio (LMA and DFD)/DFD, reaching only half of it, i.e., approximately 40%. Moreover, the ratio (LMA and DFD)/LMA is not so much lower in younger patients (between 20 and 50 years of age) compared with older age groups, as it was characteristic for diabetic patients (see Figure 3C). Figure 3D shows that female patients aged 20 to 50 years were more likely to take DFD in addition to LMA compared with men, again suggesting an association with gestational diabetes. In patients aged 50 to 80 years, there is an overall increasing trend in the ratio (LMA and DFD)/LMA, with the peculiarity that male patients with dyslipidemia were more likely to take both LMA and DFD. In patients older than 80 years, the ratio (LMA and DFD)/LMA decreases, similar to that in Figure 3C. Because female patients are much more numerous in this age group over 80 years, the trend for women (red line) in both Figure 3C,D approaches the trend for the total population (black line).

### 3.3. Laboratory Results in the Mirror of Medication

In this section, we compare some characteristic features observed in the CHDAD dataset and the HIIS dataset in relation to serum cholesterol and serum glucose levels, on the one hand, and lipid regulation medication and diabetes medication, on the other. In Section 3.1, we performed a detailed analysis of age- and sex-specific measurements of serum Gluc, Chol, LDL, HDL, and TG. Section 3.2 addresses age- and sex-specific use of medications to regulate serum lipid and/or glucose levels. Because CHDAD and HIIS are different databases, although both include cohorts in Slovenia, we cannot directly link the results. Therefore, we limit ourselves to highlighting the most striking associations between laboratory results and medication use. The results are presented in Figure 4.

In Figure 4A, we can observe three qualitatively different dynamic regimes in terms of the proportion of individuals with elevated serum glucose and cholesterol levels. The first regime, up to the age of 35–39 years for cholesterol and 55–59 years for glucose, can be approximated by power-law dynamics. In the next dynamic regime, in the age groups between 40–44 and 50–54 years for cholesterol and between 60–64 and 70–74 years for glucose, the proportion increases linearly with age, and finally, in the last tercile, the values decrease (see Figure 4A). The transition from power-law to linear dynamics is not easily explained, but could also be related to medication use. The drug intake data are shown in Figure 4B. The maximum consumption of LMA and DFD occurred in the age group 75–79 years. Interestingly, the peaks of the proportion of DFD intake and patients with elevated blood glucose are in the same age group. However, this is not the case for serum cholesterol levels and LMA.

We also calculated the rate at which medication use changed between two consecutive age groups. The results are shown in Figure 4C. For LMA, it can be seen that the peak in the proportion of patients with elevated serum cholesterol levels coincides with the highest rate of increase in LMA use, which is characteristic of the 55–59 age group. From this age group, the rate of increase in the use of LMA began to decrease. At the same time, the proportion of patients with higher than normal serum lipid levels also began to decline. This may suggest that in the younger age group, before the age of 55–59 years, the success of introducing the LMA may be related to the transition from the power-law fit to the linear fit in the proportion of patients with elevated serum cholesterol levels, among many other factors. It is difficult to explain why the peaks in the proportion of medication use and patients with elevated serum levels are not in the same age group, as was the case for DFD and elevated glucose levels (see Figure 4A,B). It should be remembered that the standards for the use of LMA are not as strictly defined as for DFD; therefore, the LMA control target is much more dependent on other factors such as age and other diseases, especially if the patient is at a higher risk for cardiovascular disease. Moreover, in the experience of Slovenian physicians, younger patients before the age of 55–59 years often refuse drug treatment with LMA because they still believe that they can regulate their lipid levels through diet, exercise, and a healthier lifestyle in general and avoid the risk of possible side effects of the drugs.

Finally, from age 75–79 years, we observed a negative rate for both LMA and DUD prescriptions (see Figure 4C). These results may suggest that older age groups, after age 75–79 years, are less likely to use medications to regulate serum lipids and blood glucose levels. Of course, this does not necessarily or at all mean that people become healthier as they age, but it is more likely that patients who take LMA and/or DFD over a long period of time, and start earlier in life, are more likely to die earlier and those who are naturally healthy are more likely to survive. To explore this question, we also take a look at the dynamics of mortality in the next section.

### 3.4. Mortality Dynamics in Association with Drug Use

Mortality data were obtained from the Statistical Office of the Republic of Slovenia [58]. Life expectancy in Slovenia is 78 years for men and 84 years for women [65]. The dataset contains the total number of deaths in a given age group, without information on the cause of death. Figure 5 shows the number of deaths within an age group with the best fit of the data. Interestingly, an inflection point is observed at the age of 65–69 years.

In Figure 5, we visually identified two dynamic regimes. The first regime includes individuals between the ages of 20 and 69. Accordingly, the second regime includes individuals older than 70. Interestingly, the age at which these two regimes are separated coincides with increased medication use (see Figure 4C and Figure 5). Hypothetically, this may suggest that medications can help shift the power-law curve fit to the right and significantly reduce mortality rates in individuals older than 65 years (see the inflection point at age 65–59 years in Figure 5). Of course, taking medications cannot extend life expectancy to infinity. Therefore, in Figure 5, actual deaths (orange curve) meet the speculated number of deaths if no or very few medications were used to treat age-related diseases (red dashed line) at around age 85.

## 4. Discussion

The aim of this study was to gain new insights into the age-related dynamics of metabolic disorders, especially dyslipidemia and hyperglycemia. Laboratory data, particularly serum lipid and glucose data, were mirrored in the context of medication use and mortality. Three different datasets were used for this purpose: first, a database of laboratory data; second, a drug database; and third, mortality data. The laboratory database with 506,083 records from 2008 to 2019 served as the basis for our analysis. The results from this dataset were interpreted in relation to 1,632,441 records of drug use by different patients in Slovenia between 2013 and 2018. Our results show that the highest percentage of individuals with hyperglycemia in the population was observed about 20 years after the peak of the percentage of individuals with hyperlipidemia. An inflection point was observed in the dynamics of lipids and glucose between the ages of 55 and 59. Interestingly, this inflection point corresponds to the greatest increase in lipid-lowering medications. At this time, the proportion of patients with elevated serum cholesterol, LDL, and TG levels began to decline, as did the corresponding mean serum levels. Our data do not allow us to establish a causal relationship between the dynamics of serum levels and medication use; however, they are an indication that medications may have an important impact on laboratory measurements, which is the goal of drug treatments, namely to regulate lipids and blood glucose levels. The second inflection point correlates with a steeper mortality rate after age 75 to 79, which significantly alters laboratory results, and in some ways counterintuitively, because it appears that the laboratory values are getting better. We suspect that if more sick people die and others stay healthier, the statistics might improve, at least in part. 

The results presented here are consistent with some previously published observations. Serum glucose levels have been found to increase with age [52,66,67]. After age 20, serum LDL levels increase markedly in both men and women. At ages 50 to 60 years (men) and 60 to 70 years (women), serum LDL levels remain at a plateau. Women have lower total cholesterol levels than men throughout their lives, but levels increase sharply after menopause and are higher than in men at age > 60 years [68]. In addition, women have higher LDL levels, while men have higher TG [69]. Feng et al. [38] reported a linear increase in serum LDL levels until about 52 years of age and a subsequent decrease in serum LDL levels. This behavior was also shown by Schubert et al. [39], Wang et al. [40], Marhoum et al. [41], Kreisberg et al. [70], and Ettinger et al. [71]. As for the sex- and age-specific values of serum HDL levels, female patients were found to have higher serum HDL levels compared with male patients, regardless of age [40,41,68,72]. In agreement with our results, Yi et al. reached similar conclusions [52]. The authors analyzed blood test results from 12.5 million Korean adult participants and found that men had higher mean serum glucose levels than women up to the age of 73 years; after this age, women had slightly higher mean serum glucose levels. In addition, the prevalence of impaired fasting glycemia (IFG) and T2DM increases with age. It is estimated that in 2005–2008, the proportion of diagnosed or undiagnosed diabetes in persons over 19 years of age in the United States increased with age. The highest proportion was estimated for persons in the age group ≥ 65 years, whereas in the age groups 20–44 years and 45–64 years, the estimated proportions were 3.7% and 13.7%, respectively [73]. A similar age-specific trend was found for male and female patients in England. The highest prevalence of diabetes was found in the 65–74 years age group, with a proportion of 15.7% in men and 10.4% in women [74].

The main limitation of our study is that it is a population-based observational study, so we could not analyze causality at the level of the individual patient. We could only observe a snapshot of population data and cannot perform a longitudinal analysis of individuals. We lack data on causal relationships, and we cannot say, for example, that an individual first developed dyslipidemia and then hyperlipidemia. Furthermore, we do not have data on diabetes diagnoses, so the observed hyperglycemia can only serve as an indirect indication of diabetes, because we know that a high blood glucose level is the main inherent criterion for the diagnosis of diabetes mellitus, either by direct measurements of serum glucose levels or by surrogate values such as HbA1c (see the Supplement for a detailed description of diagnostic criteria for T2DM). The use of HbA1c levels is even more common and promising; however, we did not have these data for HbA1c until recent years, so we could not use this parameter in our analysis. In addition, the CHDAD data were limited to one region in Slovenia—the Podravska region. However, we believe that the results can be generalized due to genetic coherence in Slovenia. Only sex and age can be considered as influencing factors. In addition, some possible confounding factors such as pubertal stage, physical activity, dietary habits, and socioeconomic status were not considered. In addition, we could not link the medication intake data (HIIS) to the CHDAD dataset for the same patients because of legal restrictions. Therefore, we cannot conclusively answer the question of the extent to which the decreasing proportion of patients with abnormal serum lipid and/or glucose levels is attributable to drug treatment. Additional data are needed to answer this question more precisely. However, our results clearly suggest that the decreasing proportion of the population with abnormal lipid and/or glucose levels may be influenced by medication use, and, at older ages, by mortality.

In future studies, if sufficient data are available, the size of LDL particles should be considered in addition to the measured serum LDL concentration. Elderly people and people with T2DM typically have smaller and more oxidized LDL particles [75]. The small LDL particles oxidize faster, are cytotoxic, and pose an increased risk to metabolic health. Therefore, LDL particle size could serve as an additional indicator of early metabolic disorders. Additional data increase the complexity of the analysis, but more data, especially linked data, are needed to provide better research results on associations between metabolic and other disorders. In this light, the era of machine learning and data mining can be an efficient tool to achieve more accurate patient-specific risk factor assessment and diagnosis [76,77], disease risk prediction [78,79,80,81], and appropriate drug selection for the patient’s specific disease [82,83]. In the context of personalized medicine, patient-specific treatment should not only consider and evaluate average values [84]. This combination of various patient-specific data can potentially greatly improve the early prediction of T2DM based on laboratory measurements.

## 5. Conclusions

Our results show that metabolic disorders in the population are manifested first by elevated lipid levels and then by elevated glucose levels in individuals. The time interval between the highest percentage of individuals with hyperlipidemia and the highest percentage of individuals with hyperglycemia is approximately 20 years. Great care must be taken when analyzing laboratory data, which includes serum measurements, because laboratory data are influenced by drug use. Therefore, this must be taken into account. In addition, the proportion of patients with hyperlipidemia or/and hyperglycemia is influenced by mortality, especially in older age groups. Of course, mortality does not directly affect the data measured in the laboratory, as is the case with drug use, but mortality does affect the statistics, as healthier subjects have a higher probability of survival. Our study provides only an indication of this problem. However, more detailed studies of the interaction of serum lipid and glucose dynamics will be needed in the future, requiring more complex data in conjunction with other databases (e.g., prescribed medications, time to diagnosis, clinical treatments). This will lead to much more complex analyses with a big data pool, involving new methods of data mining and artificial intelligence, opening the door to broader implementation of personalized medicine.

## Figures and Tables

**Figure 1 jpm-12-00280-f001:**
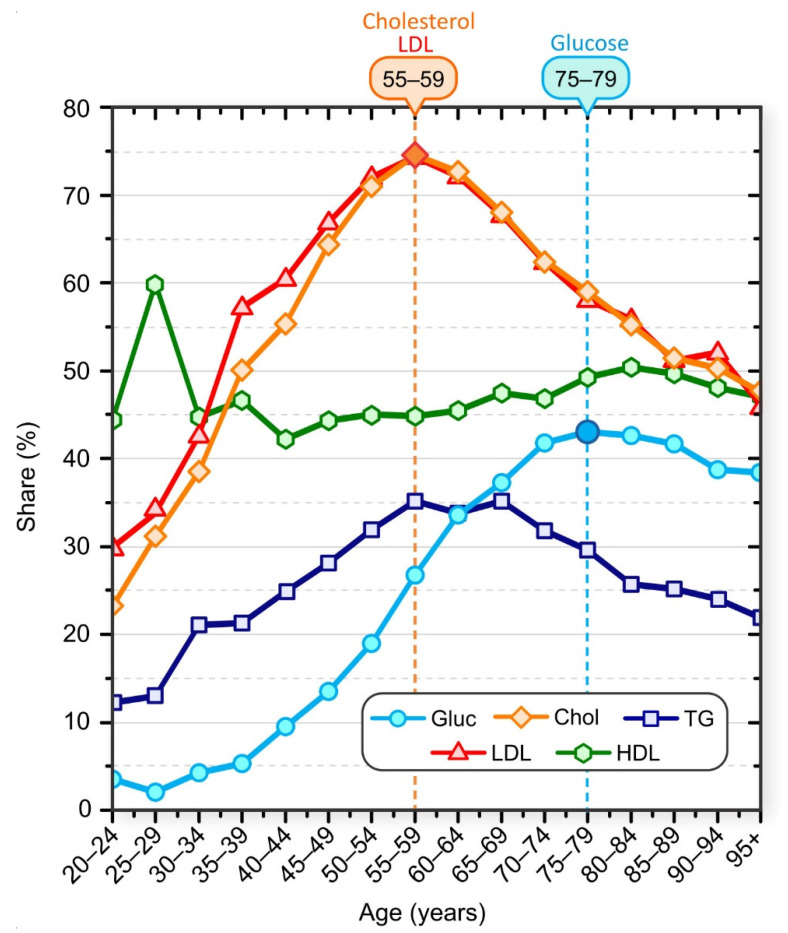
Share of patients with elevated laboratory test results: Cholesterol (orange diamonds), LDL (red triangles), glucose (blue circles), HDL (green hexagons), and TG (dark blue squares). Patients were aggregated into nonoverlapping 5-year age groups.

**Figure 2 jpm-12-00280-f002:**
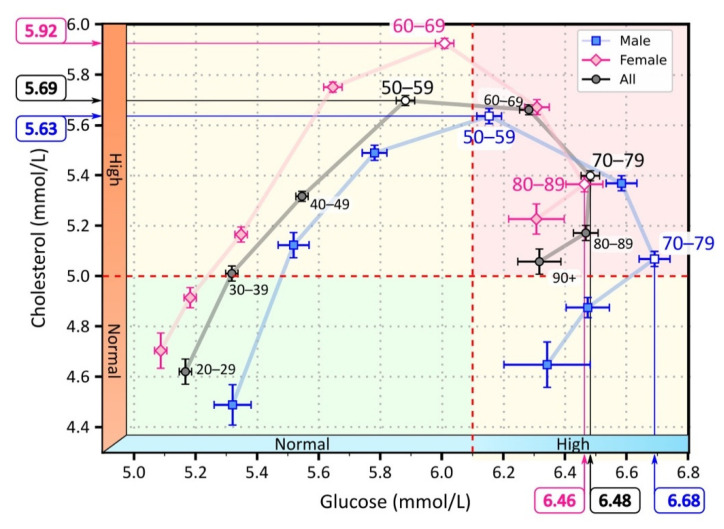
Mean values of serum cholesterol and glucose for different age groups. The light pink diamonds represent measurements for female patients, the light blue squares for male patients, and the black circles represent the overall mean for male and female patients. The white-filled symbols indicate extreme values: the maximum cholesterol values in the age groups 50–59 (men), 60–69 (women), and 50–59 (all), and the maximum glucose values in the age groups 70–79 (men), 80–89 (women), 80–89 (all). The background colors in the figure highlight the area with normal cholesterol and glucose levels (green), the area with elevated cholesterol or glucose levels (yellow), and the area with elevated glucose and cholesterol levels (red).

**Figure 3 jpm-12-00280-f003:**
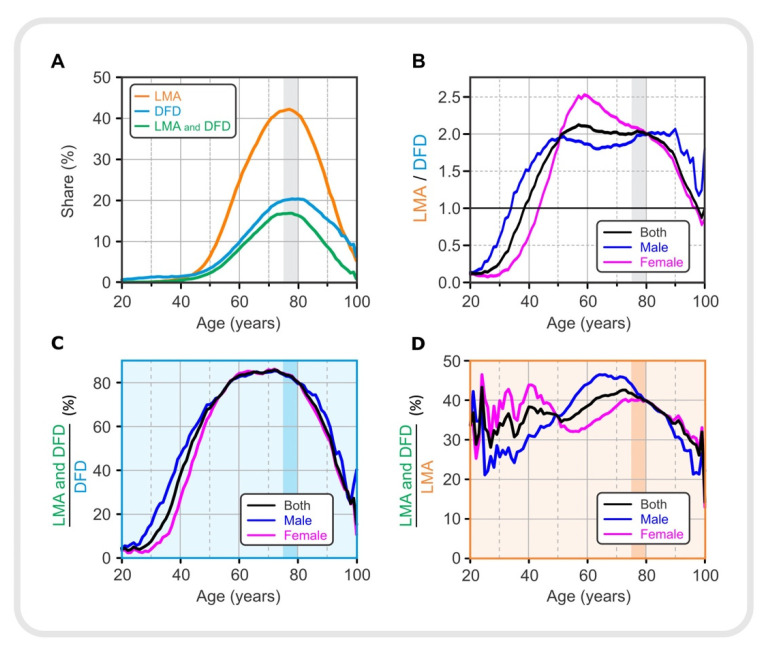
Drug use in different age groups. (**A**) Proportion of lipid-modifying agents, LMA, (orange), drugs for diabetes, DFD, (light blue), LMA, and both LMA and DFD (green). (**B**) The ratio between the proportions of LMA and DFD. (**C**) Proportion of patients taking both types of medication (LMA and DFD) normalized to diabetic patients taking DFD in the same age and sex group. (**D**) Proportion of patients taking both types of medication (LMA and DFD) normalized to patients with lipid dysregulation taking LMA in the same age and sex group. LMA—lipid-modifying agents, DFD—drugs for diabetes.

**Figure 4 jpm-12-00280-f004:**
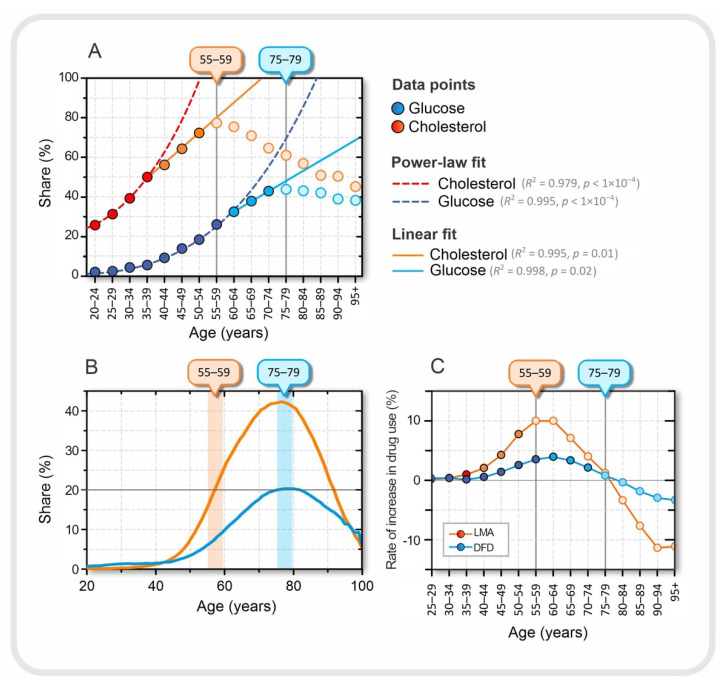
Relationship between laboratory results and drug use. (**A**) Proportion of patients in a given age group with either elevated serum lipids (orange) or elevated serum glucose (blue), approximated with best fit, a power-law fit up to ages 35–39 for cholesterol and 55–59 for glucose. A good linear fit is observed in the age groups between 40–44 and 50–54 for cholesterol and between 60–64 and 70–74 for glucose. The dark orange and dark blue dots were used for the power-law fit, the orange and blue dots for the linear fit, and the light blue and light orange dots represent the decreasing trend of the proportions in the last tercile. The age at which the proportion of patients with elevated serum lipid and serum glucose levels peaks is indicated by the orange and blue boxes on the upper horizontal axes, respectively. (**B**) Proportion of patients of a given age taking lipid-modifying agents, LMA (orange) and drugs for diabetes, DFD, (blue). (**C**) Rate of increase in the proportion of patients between the two consecutive age groups taking a particular type of medication.

**Figure 5 jpm-12-00280-f005:**
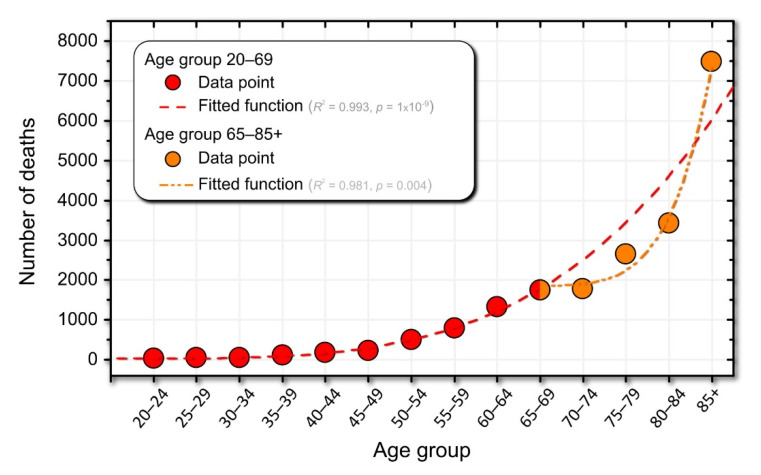
The number of deaths in Slovenia (data from the Statistical Office of the Republic of Slovenia)**.** The dashed red line represents the power-law fit of the data. The dash-dotted orange curve represents the power-law fit for the population over 70 years of age). The legend contains information on the best-fit functions.

## Data Availability

Data supporting the findings of this study were provided by the Community Healthcare Center Dr. Adolf Drolc in Maribor, Slovenia, and the Health Insurance Institute of Slovenia. Restrictions apply to the availability of the data used in this study; therefore, the data are not publicly available. However, the derived data supporting the findings of this study are available upon request from the corresponding authors, M.M. and J.Z.

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
