# Peer review of "Age-Related Changes in Lipid and Glucose Levels Associated with Drug Use and Mortality: An Observational Study"

_jpm, 2022, doi:10.3390/jpm12020280_

Round 1
Reviewer 1 Report
In this manuscript, the authors report an observational study in the context of age-related changes in lipid and glucose levels associated with drug use and mortality. The subject is of interest for the journal readers, the used methods seem to be adequate, results are well discussed and the conclusions consistent with the obtained data. The work is very interesting and after the small changes, I suggest that can be published.
- The quality of each figure has to be improved.
-They should run spell-check and carefully check for typos.
Reviewer 2 Report
Although the authors say that they cannot prove causality in this study, they say that dyslipidemia occurs first and then diabetes occurs about 20 years later. Unfortunately, the explanation in this paper is not convincing enough to support the hypothesis.
Line 164.
According to the author, patient records cover 11.8% of the population, but doesn't this mean the total number of people? If we are talking about coverage, shouldn't we be talking about the actual number of people?
“It should be noted that the total patient records in the CHDAD dataset cover approximately 11.8% of the population of Slovenia. “
Glucose cannot necessarily diagnose diabetes using blood glucose at any time only once, why did the author use glucose levels, why did the author not use HbA1c?
Line 484.
The author says that the administration of medication affected the change in inflection point, but is there a national index or guideline for starting treatment at what level of blood glucose?
Isn't HbA1c the index for starting treatment?
In Fig. 1, why is the administration rate of LMA so low when the percentage of patients with dyslipidemia is about 75%? Is there a uniform standard for lipids, or is each physician administering the drug individually?
In principle, insulin is the only drug that can be used for gestational diabetes, but does this context mean that DFD increases because of the transition to type 2 diabetes after gestational diabetes? I think this point should be stated more clearly.
Line 439.
As an explanation for the hypothesis, could it be because the LMA control target became lower as age increased and ACS increased? In other words, with age, the number of complications increased and the target was lowered. This may explain why LMA administration is decreasing in spite of the high prevalence of dyslipidemia in line 431.
“Hypothetically, the transition from the power-law fit to the linear fit in the proportion of patients with elevated serum cholesterol levels could be related to the steepest increase in the prescription of LMA in patients older than 40 years. “
Fig. 4C shows that LMA tends to be discontinued at older ages. On the other hand, Fig. 4D shows that the decrease in DFD is not so large. Rather than a decrease due to death, this may be due to abandonment of LMA inhibition of complications due to old age. If deaths increase due to vascular complications from diabetes and dyslipidemia, then the number of people receiving DFD should decrease as well.
Reviewer 3 Report
I appreciate the hard work that writers have put into performing this study but the final report should be improved.
The abstract should be rewritten it does not follow a logical style. State your purpose first. Then give details of methodology then state the results and then give a conclusion.
Please shorten the in introduction section and focus more on previous similar studies which have tried to establish a link between hyperlipidemia and diabetes.
The last sentence of introduction should give the reason for conducting the present study and the main objectives of the study.
Some sentences in introduction should be reconsidered and amended.
In methods section give more details about the accuracy of your data and how reliable you think your data is.
In results section tell us what percentage of data was missing or thrown out because of errors in data collection or other reasons.
In introduction section you say your main goal is to the relationship between dyslipidemia, hyperglycemia, and later diabetes but you somehow miss this goal in the methodology and result section and consequently in discussion section.
The easiest way to study the relation between early hyperglycemia and later diabetes is to divide your patients to two groups group one patients with early stage hyperlipidemia and group 2 patients without early hyperglycemia and then compare these two groups regarding the prevalence of later hyperglycemia and diabetes.
Focusing too much on drug usage when the data regarding drug usage comes from two different databases has weakened your article.
Amend your discussion after providing more results related to your stated goal as I have suggested above.
Summarize your conclusion to one paragraph. Do not repeat unnecessary sentences in conclusion section.
Round 2
Reviewer 2 Report
I would like to ask the author in response to their answer below. Does DFD include insulin since we consider insulin to be the only diabetes drug associated with type 1 diabetes? The author should mention the definition of DFD and what it includes, such as SU and DPP4 inhibitors. At the same time, the definition of LMA (statin? fibrate?) should be mentioned. “Younger patients were also found to have a higher proportion of DFD compared to LMA (see Figure 3B), which could be associated with type 1 diabetes.”Author Response
We thank the reviewer for pointing this out. Yes, the DFD drug group includes insulin, and the LMA includes both statins and fibrates. To clarify this for the reader, we now refer to the supporting information file in the revised version of the manuscript; we write: “For a complete list of all drugs included under LMA and DFD, with the corresponding total frequency of prescriptions, see Tables S5 and S6 in the Supplement.” In the Supplement we added a new section titled "List of DFD and LMA drugs considered " in which we list all DFD and LMA drugs with the corresponding ATC code and total frequency of prescriptions.Reviewer 3 Report
In the last sentence of introduction only mention the reason for conducting the present study and the main objectives of the study. Do not give details like the number of participants place of study, etc... summarize the paragraph.
Author Response
We thank the reviewer for this comment. In accordance with the reviewer’s recommendations, we have completely rewritten the last paragraph of the introduction; it now reads as follows:
“In the present study, we aim to integrate perspectives on lipid and glucose profiles in the context of age, sex, medication use, and mortality. First, we analyze laboratory data to find evidence of dysregulations in lipid and glucose metabolism and to determine how the dynamics depend on sex and age. The focus is on whether some disorders occur earlier in life than others and to what extent this depends on sex. Second, we are investigating whether and to what extent laboratory data are affected by medication use, and third, we are interested in how the statistics of lipid and glucose measurements are affected by mortality, particularly in older age groups. The data for laboratory tests, medication use, and mortality were obtained from three separate data sets and do not allow for a longitudinal study, but the analysis allows us to clearly demonstrate that lipid and glucose profiles in a population statistic are qualitatively altered by medication treatment and mortality. This underscores the importance of providing an overall picture when compiling statistics on specific laboratory data that reflect a wide range of all these parameters, including drug treatment, mortality, and, in perspective, probably many other factors such as lifestyle, dietary habits, and physical activity.”